# A Calibration Method for a Self-Rotating, Linear-Structured-Light Scanning, Three-Dimensional Reconstruction System Based on Plane Constraints

**DOI:** 10.3390/s21248359

**Published:** 2021-12-15

**Authors:** Jianping Zhao, Yong Cheng, Gen Cai, Shengbo He, Libing Liao, Guoqiang Wu, Li Yang, Chang Feng

**Affiliations:** 1Institute of Optics and Electronics, Chinese Academy of Sciences, Chengdu 610209, China; cyong@ioe.ac.cn (Y.C.); imcaigen@ioe.ac.cn (G.C.); heshengbo@ioe.ac.cn (S.H.); liaolibin@163.com (L.L.); 13102681929@163.com (G.W.); YL18349225132@163.com (L.Y.); 2University of Chinese Academy of Sciences, Beijing 100049, China

**Keywords:** linear-structured light, three-dimensional reconstruction, rotation center, calibration

## Abstract

This paper proposes a calibration method for a self-rotating, linear-structured-light (LSL) scanning, three-dimensional reconstruction system based on plane constraints. The point cloud of plane target collected by the self-rotating, LSL scanning, 3D reconstruction system should be constrained to the basic principle of the plane equation; it can quickly and accurately calibrate the position parameters between the coordinate system of the LSL module and the coordinate system of the self-rotating, LSL scanning, 3D reconstruction system. Additionally, the transformation equation could be established with the calibrated optimal position parameters. This paper obtains the above-mentioned position parameters through experiments and uses the calibrated self-rotating, LSL scanning, 3D reconstruction system to perform three-dimensional scanning and reconstruction of the test piece. The experimental results show that the calibration method can effectively improve the measurement accuracy of the system.

## 1. Introduction

A self-rotating, linear-structured-light (LSL) scanning, 3D reconstruction system mounted on a mobile robot can be applied to the 3D reconstruction of the internal dimensions of structures such as pipelines [1,2,3], tunnels [4], tanks, indoor environments [5,6,7], nuclear reactor internals [8,9], underwater environments [10], etc. That can help identify internal defects, damages, leakages, etc. As shown in Figure 1, the self-rotating, LSL scanning, 3D reconstruction system is composed of an LSL module and a rotating module. The coordinate system of the self-rotating, LSL scanning, 3D reconstruction system is usually established on the rotation center of the rotating module. However, the position parameters between the coordinate system of the LSL module and the rotation center of the rotating module are difficult to determine by the mechanical installation position. Inaccurate position parameters will lead to a decrease in system measurement accuracy [11]. Therefore, it is necessary to establish a set of calibration methods to calibrate the position parameters between the coordinate system of the LSL module and the coordinate system of the self-rotating, LSL scanning, 3D reconstruction system to improve the measurement accuracy of the system.

Scholars have proposed some calibration methods for solving the transformation matrix between the measurement coordinate system and the world coordinate system. However, most of them are the calibration methods for rotating 2D light detection and ranging (LIDAR). Zeng et al. [12] established the measurement coordinate system when the 2D LIDAR rotation angle is zero. When installing, make each axis parallel to each axis of the motion module coordinate system as much as possible. The origins of the two coordinates coincide in the vertical direction. Since it is difficult to ensure that the coordinate axis is completely parallel, a calibration method for the deflection angle of the center line and the coordinate axis was established. They scanned a calibration plane that is perpendicular to the Z axis of the measurement coordinate system and divided the scanned data into two, then used the projection of the measured distance of each point on the plane to the Z axis. The distances are equal; the first data are used to find the offset angle, and the second data are used to verify the results, and finally, the optimal solution is obtained. However, this method only considers the skew problem and does not calibrate the position parameters of the two coordinate origins. Yu Qiqi et al. [13] obtained the distance between the origins of the two coordinates directly through the installation and only calibrated the angle between the lines of the two origins with respect to the horizontal plane through the parallel planes at different distances from the device. Therefore, the direct position parameters of the origins of the two coordinate systems are not fully calibrated.

Alismail et al. [14] proposed an algorithm to automatically calibrate the rotation center for the problem that the 2D LIDAR rotation center and the driving device center may not overlap, and this method does not require a special target and can be used in general measurement scenarios. This method is based on the assumption that the surface in the local neighborhood of the 2D LIDAR point cloud can be well approximated to a plane. Since the full scan is symmetrical, the full scan is divided into two half scans of the same object, and the distance from the plane to the center in the range of the corresponding angle laser point is equal, and the transformation matrix that minimizes the dissimilarity of the two data is calculated so that the calibration of the center of rotation is completed. This method does not require special calibration objects, but the algorithm is more complicated. Huang Fengshan et al. [15] used a rectangular parallelepiped with a triangular prism in the middle channel as the calibration piece to complete the calibration. The calibration piece is large and requires special customization. Cai et al. [16] proposed using a special calibration plate with hollow holes for calibration, This method needs to fit the coordinates of each center according to the measured data and then complete the calibration according to the position relationship of each center. The calibration methods for rotating 2D LIDAR can be used as references for calibrating self-rotating, LSL scanning, 3D reconstruction systems, but they cannot be directly applied.

Li and Xi [17] and Xiao et al. [18] presented a rotational laser scanner by mounting the LSL sensor to a turntable, but they did not mention the calibration of the center of rotation. Manakov et al. [19], Wissel et al. [20], and chi et al. [10] designed a galvanometric laser scanner with a laser projector and proposed a model-driven calibration method, but they all assumed that the galvanometer rotation axis coincides with the line intersected by the mirror of the galvanometer and the laser plane. Yang M et al. [21] proposed a flexible plane-constraint-based calibration method for the galvanometric laser scanner, which is effective in the vision inspection system, but this method needs to be combined with a lookup table.

Aiming at the above problems, this paper proposes a calibration method for a self-rotating, LSL scanning, 3D reconstruction system based on plane constraints. By collecting the point cloud data of the plane target, based on the basic principle that the points on the plane target should be constrained by the plane equation, position parameters between the coordinate system of the LSL module and the coordinate system of the self-rotating, LSL scanning, 3D reconstruction system were calculated.

## 2. Transformation between Coordinate Systems

In the establishment of the coordinate system of a self-rotating, LSL scanning, 3D reconstruction system, $O_{r}X_{r}Y_{r}Z_{r}$, the axis $Z_{r}$ is coincident with the rotation axis of a high-precision rotating turntable, and the direction is upward. The plane $X_{r}O_{r}Y_{r}$ is established based on the right-hand rule. Additionally, the plane $X_{r}O_{r}Y_{r}$ and the plane of the coordinate system of the LSL module are set coplanar when the rotation angle is zero, while the axis $Y_{r}$ is parallel to the axis Y and the axis $X_{r}$ is parallel to the X axis. As shown in Figure 2, $O_{r}X_{r}Y_{r}Z_{r}$ is the coordinate system of the self-rotating, LSL scanning, 3D reconstruction system, while $O_{r}X_{r}Y_{r}Z_{r}$ is the coordinate system of the high-precision rotating turntable. OXYZ is the coordinate system of the LSL module when the rotation angle is zero. $O_{i}X_{i}Y_{i}Z_{i}$ is the coordinate system of the LSL module when the axis rotates clockwise by an angle α.

Point P is a measurement point on the laser plane when the rotation angle α of the self-rotating, LSL scanning, 3D reconstruction system. Since the laser plane coincides with the plane $Y_{i}O_{i}Z_{i}$, the coordinate of point P in the coordinate system is $\left(0,y_{i},z_{i}\right)$. The coordinate of the point P in the coordinate system $O_{r}X_{r}Y_{r}Z_{r}$ is $\left(x_{r},y_{r},z_{r}\right)$, since the Z axes of the two coordinate systems are parallel and in the same direction, so $z_{r}=z_{i}$.

When the rotation angle of the self-rotating, LSL scanning, 3D reconstruction system is zero, the coordinate value of the origin O of the LSL module in the coordinate system $O_{r}X_{r}Y_{r}Z_{r}$ be (M,N,0). Pass point O to make a vertical line perpendicular to the axis $X_{r}$, and point A is the vertical foot. When the rotation angle of the self-rotating, LSL scanning, 3D reconstruction system is α, the origin of the coordinate system of the LSL module shifts from point O to point $O_{i}$, and the corresponding position of point A shifts to point Ai.

From the geometric relationship,
(1)$$|O_{i}A_{i}|=|OA|=\left|N\right|$$
(2)$$|O_{r}A_{i}|=|O_{r}A|=\left|M\right|$$
where $O_{r}P^{’}=R$, $P^{’}A_{i}=L$, $∠O_{r}P^{’}A_{i}=\beta ,\;∠AO_{r}P^{’}=\gamma$. According to the geometric relationship in Figure 2, we can obtain
(3)$$L=\left|N+y_{i}\right|$$
(4)$$R=\sqrt{M^{2}+L^{2}}$$
(5)$$\beta =\arctan\frac{\left|M\right|}{L}$$
(6)$$\gamma =\alpha -\beta +\frac{\pi }{2}$$

Therefore, the coordinate value $\left(x_{r},y_{r},z_{r}\right)$ of the point P in the coordinate system $O_{r}X_{r}Y_{r}Z_{r}$ can be expressed as follows [18]:(7)$$\left\{ \begin{array}{l} x_{r}=R\times \cos\left(-\gamma \right)=\sqrt{M^{2}+{\left(N+y_{i}\right)}^{2}}\times \cos\gamma =\sqrt{M^{2}+{\left(N+y_{i}\right)}^{2}}\times \sin\left(\arctan\frac{\left|M\right|}{\left|N+y_{i}\right|}-\alpha \right)\\ y_{r}=R\times \sin\left(-\gamma \right)=-\sqrt{M^{2}+{\left(N+y_{i}\right)}^{2}}\times \cos\left(\alpha -\arctan\frac{\left|M\right|}{\left|N+y_{i}\right|}\right)\\ z_{r}=z_{i} \end{array}\right.$$

It can be seen from Equation (7) that the coordinate value $\left(x_{r},y_{r},z_{r}\right)$ of the point P in the coordinate system $O_{r}X_{r}Y_{r}Z_{r}$ is related to the parameters M, N, α and the coordinate value $\left(0,y_{i},z_{i}\right)$ of the point P in the coordinate system $O_{i}X_{i}Y_{i}Z_{i}$. The angle α can be measured by the encoder and the coordinate value $\left(0,y_{i},z_{i}\right)$ of the point P in the coordinate system $O_{i}X_{i}Y_{i}Z_{i}$ can be obtained by the LSL module, while the position parameters M and N can only be a rough estimate of their value range, and further calibration is required to obtain their precise values.

## 3. Calibration Based on Plane Constraints

This paper proposes a calibration method for self-rotating, LSL scanning, 3D reconstruction systems based on plane constraints to achieve the precise acquisition of the position parameters M and N described in Section 2. The specific method is as follows:

(1) First, according to the design parameters and installation positions of the LSL module and high-precision rotating turntable, the position parameters are preliminarily estimated a ≤ M ≤ b, c ≤ N ≤ d.

(2) As shown in Figure 3, the calibration plane is placed within the measurement range of the self-rotating, LSL scanning, 3D reconstruction system. The high-precision rotary turntable is rotated so that the LSL module sweeps across the plane, and at the same time, the data of the plane profile are collected in the LSL module at various angles; then, the transformation method of Equation (7) is used to unify the point cloud data to the coordinate system of the self-rotating, LSL scanning, 3D reconstruction system.

Assume that the three-dimensional coordinates of the measurement point on the calibration plane in the self-rotating, LSL scanning, 3D reconstruction system are $\left(x_{1},y_{1},z_{1}\right)$, $\left(x_{2},y_{2},z_{2}\right)$……$\left(x_{n},y_{n}^{},z_{n}\right)$. These measurement points are all located on the same plane, so these points are constrained to a plane. The plane equation is Z=AX+BY+C, and there is
(8)$$\left\{ \begin{array}{l} Ax_{1}+By_{1}+C=z_{1}\\ Ax_{2}+By_{2}+C=z_{2}\\ \ldots \\ Ax_{n}+By_{n}+C=z_{n} \end{array}\right.$$
which is
(9)$$\left(\begin{array}{l} \begin{array}{ccc} x_{1} & y_{1} & 1\\ x_{2} & y_{2} & 1\\ \ldots & & \end{array}\\ \begin{array}{ccc} x_{n} & y_{n} & 1 \end{array} \end{array}\right)\left(\begin{array}{c} A\\ B\\ C \end{array}\right)=\left(\begin{array}{c} z_{1}\\ z_{2}\\ \ldots \\ z_{n} \end{array}\right)$$

Set
(10)$$\left(\begin{array}{l} \begin{array}{ccc} x_{1} & y_{1} & 1\\ x_{2} & y_{2} & 1\\ \ldots & & \end{array}\\ \begin{array}{ccc} x_{n}^{} & y_{n} & 1 \end{array} \end{array}\right)=D$$
(11)$${\left(\begin{array}{ccc} A & B & C \end{array}\right)}^{T}=\phi$$
(12)$${\left(\begin{array}{cccc} z_{1} & z_{2} & \ldots & z_{n} \end{array}\right)}^{T}=\delta$$

Solve the equation Dϕ=δ and obtain the fitted plane by the least square method, the least-square solution of the equation is
(13)$$\phi ={\left(D^{T}D\right)}^{-1}D^{T}\delta$$

The distance from the points to the fitting plane is $d_{i}^{}$, where $d_{i}^{}=\frac{\left|AX_{i}+BY_{i}-Z_{i}+C\right|}{\sqrt{A^{2}+B^{2}+1}}$, so the root-mean-squared error of the distance from the measuring point to the fitting plane is
(14)$$rmse_{i}=\sqrt{\frac{1}{n}\sum_{i=1}^{n}d_{i}^{2}}$$

Change the calibration plane position, as shown in Figure 4, and collect the point cloud of the planes at different positions. According to the above calculation method, we can obtain the root-mean-squared error of the fitting plane at different positions $rmse_{1}$, $rmse_{2}$……$rmse_{m}$. The root-mean-squared error of each fitting plane and the average value is
(15)$$S=\frac{1}{m}\sum_{i=1}^{m}rmse_{i}$$

Take Equation (15) as the objective function and search for the best value of M and N in the interval of a ≤ M ≤ b, c ≤ N ≤ d with a certain step length. Minimize the value of S, and use this optimized value of M and N as the result of positional parameter calibration. The Algorithm 1 is as the follows:
**Algorithm 1:** Calibration of self-rotating, LSL scanning, 3D reconstruction system based on plane constraints.Input:1. The collected point cloud data of m groups of planes that are not parallel to each other;2. The value range of parameter M is [a, b]. The value range of parameter N is [c, d].Algorithm steps:3. M = a, N = c, S = 100, step size is δ;4. **While** M≯b **do**5. **While** N≯c **do**6. Based on the least square method to get the fitting planes, and solve the root mean square error of each group of point clouds to each fitting plane rmse1,rmse2……rmsem; 7. **If**
S>1m∑i=1mrmsei
**then**8. S=1m∑i=1mrmsei; MOPT=M; NOPT=N;9. end if10. N = N + δ;11. end while12. M = M + δ;13. end whileOutput:14. The minimum S value and position parameters MOPT and NOPT.

## 4. Experiment and Results

As shown in Figure 5, the plane of the aluminum alloy plate was placed within the measurement range of the self-rotating, LSL scanning, 3D reconstruction system, and the self-rotating, LSL scanning, 3D reconstruction system was controlled to scan and collect the point cloud data of the aluminum alloy plate. The collected data were the data in the coordinate system of the LSL module. The plane position of the aluminum alloy plate was adjusted, and the point cloud data were collected three times.

According to the self-rotating, LSL scanning, 3D reconstruction system designed in Section 2, it was preliminarily estimated that the M value would be in the interval (110, 130), and the N value would be in the interval (395, 425). Using the calibration method described in Section 3 to process the collected data, the relationship between the parameters M, N, and the root-mean-squared error S was obtained, as shown in Figure 6. The position parameters Mopt and Nopt corresponding to the minimum root-mean-squared error were the calibration results of the parameters M and N, as shown in Table 1.

In order to verify the accuracy of the parameters Mopt and Nopt, obtained in the above calibration process, the test piece, as shown in Figure 7, was processed, its characteristic dimensions were measured by a coordinate measurement machine, and the size measured by the coordinate machine was used as the true value.

In Figure 8, the test piece was placed within the measurement range of the self-rotating, LSL scanning, 3D reconstruction system, and the self-rotating, LSL scanning, 3D reconstruction system was controlled to scan and collect the point cloud of the test piece’s surface.

To convert the collected data into the data in the coordinate system of the self-rotating, LSL scanning, 3D reconstruction system, M = 123 and N = 409, as shown in Figure 9; the relevant feature size was then measured.

As shown in Table 2, when M = 123 and N = 409, where M and N are the optimal value, the measurement accuracy of each feature size on the test piece is better than others. When the value of M and N is smaller than the optimal value, the measured value of the feature size along the scanning direction on the test piece is smaller than the measured value when M and N take the optimal value, while the measured value of the feature size perpendicular to the scanning direction is equivalent when M and N take the optimal value. When the value of M and N is greater than the optimal value, the measured value of the feature size along the scanning direction of the test piece is greater than the measurement value when M and N take the optimal value, and the measured value of the feature size perpendicular to the scanning direction is equivalent when M and N take the optimal value.

## 5. Conclusions

The coordinate system of the LSL module does not coincide with the coordinate system of the self-rotating, LSL scanning, 3D reconstruction system, and the positional parameters between them are difficult to measure through mechanical installation. This paper first analyzed the transformation relationship between the coordinate system of the LSL module and the coordinate system of the self-rotating, LSL scanning, 3D reconstruction system, then proposed a calibration method for the self-rotating, LSL scanning, 3D reconstruction system based on plane constraints, and finally, designed the validation experiment. It was confirmed by the experiment that the proposed method can accurately calculate the positional parameters between the coordinate system of the LSL module and the coordinate system of the self-rotating, LSL scanning, 3D reconstruction system, and can significantly improve the measurement accuracy of the measurement system. 

## Figures and Tables

**Figure 1 sensors-21-08359-f001:**
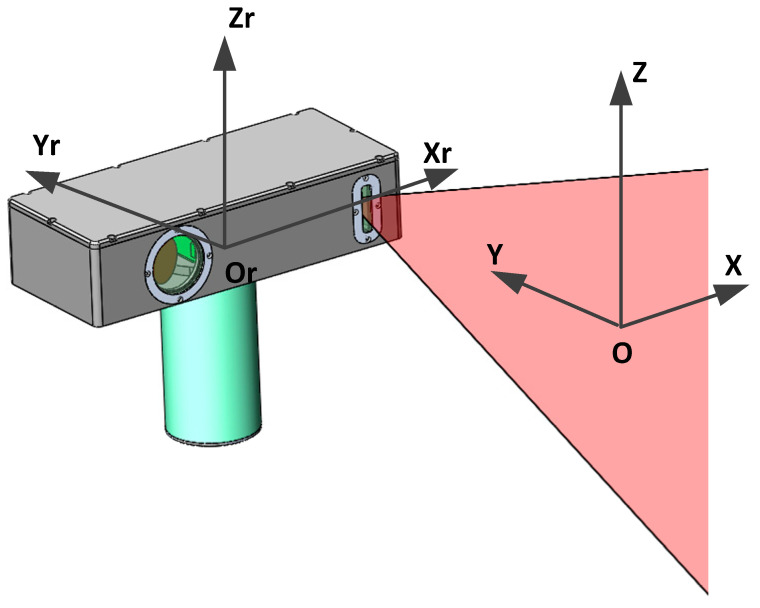
The self-rotating, linear-structured light scanning measurement system.

**Figure 2 sensors-21-08359-f002:**
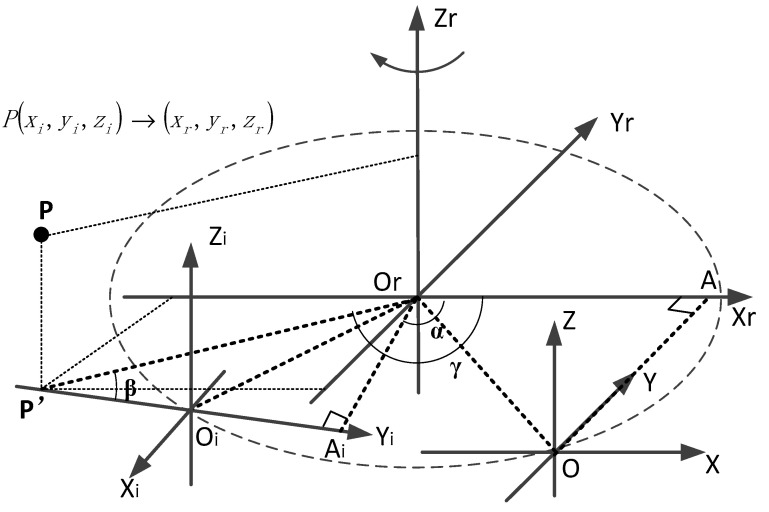
The transformation between the coordinate system of the LSL module and the coordinate system of the self-rotating, LSL scanning, 3D reconstruction system.

**Figure 3 sensors-21-08359-f003:**
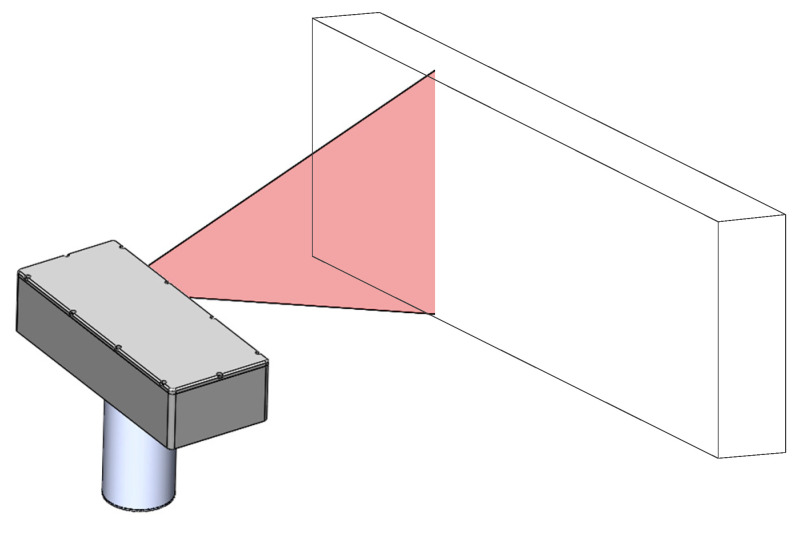
The calibration of self-rotating, LSL scanning, 3D reconstruction system based on plane constraints.

**Figure 4 sensors-21-08359-f004:**
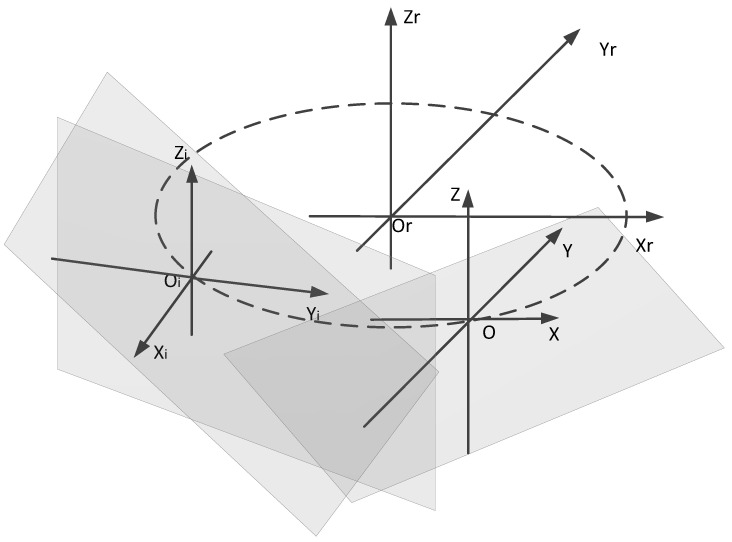
Schematic of different plane positions.

**Figure 5 sensors-21-08359-f005:**
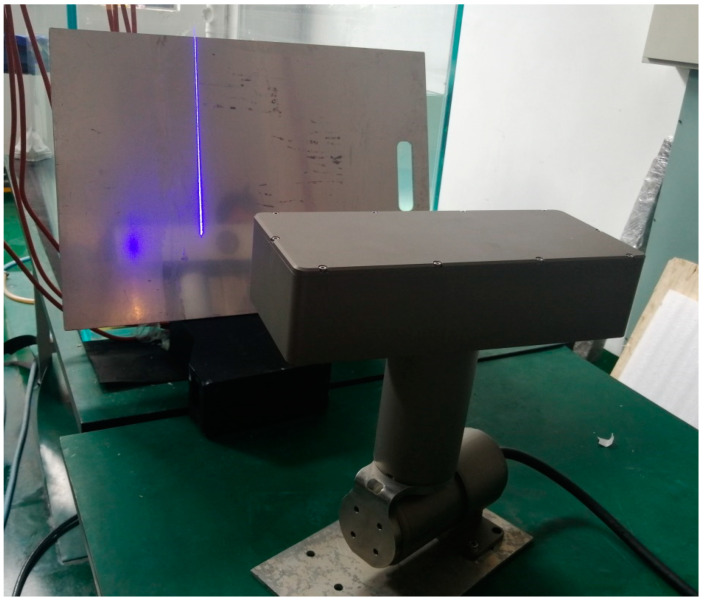
Calibration experiment of self-rotating, LSL scanning, 3D reconstruction system based on plane constraints.

**Figure 6 sensors-21-08359-f006:**
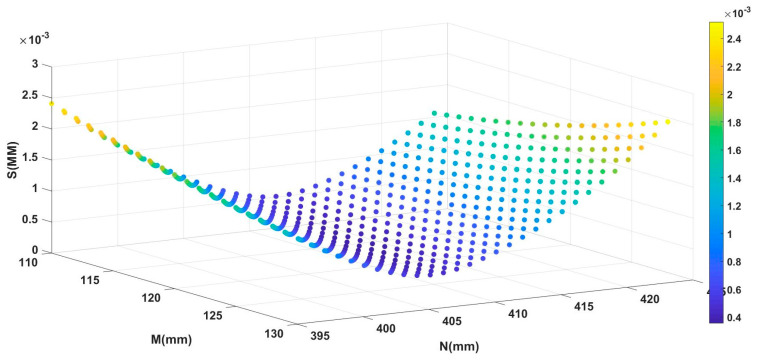
The relationship between the root-mean-squared error S value and the parameters M and N.

**Figure 7 sensors-21-08359-f007:**
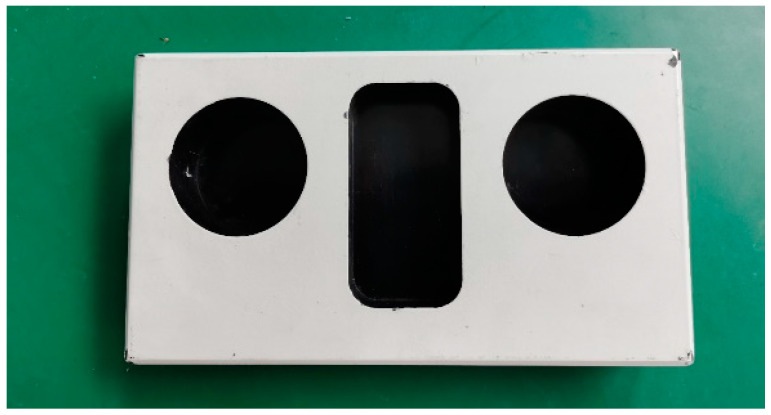
Test piece.

**Figure 8 sensors-21-08359-f008:**
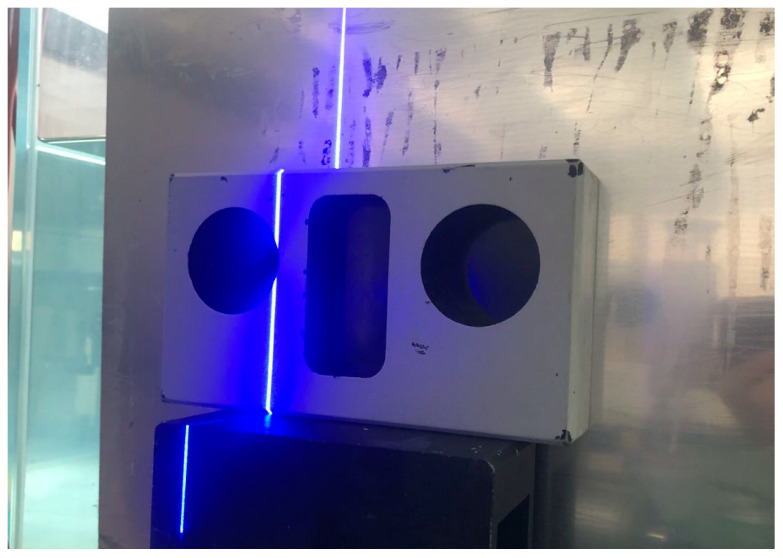
The measurement experiment of test piece.

**Figure 9 sensors-21-08359-f009:**
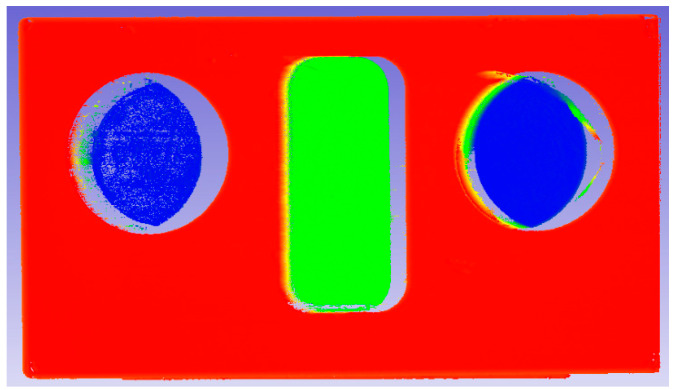
Point cloud data in the coordinate system of the self-rotating, LSL scanning, 3D reconstruction when M = 123 and N = 409.

**Table 1 sensors-21-08359-t001:** Calibration results of parameters M and N.

Mopt	Nopt	Min(S)
123 mm	409 mm	5.88 × 10^−4^ mm

**Table 2 sensors-21-08359-t002:** Comparison of measurement results.

Measurement Target		Values Measuredby CMM	Values and Errors Measured by LSL When M = 123, N = 409 (10 Times)		Values and Errors Measured by LSL When M = 110, N = 395 (10 Times)		Values and Errors Measured by LSL When M = 130 and N = 425 (10 Times)	
Left round groove	Diameter	50.01 mm	49.97 mm	−0.08%	47.24 mm	−5.54%	51.58 mm	+3.14%
	Depth	30.03 mm	29.96 mm	−0.23%	29.94 mm	−0.30%	29.86 mm	−0.57%
Right round groove	Diameter	50.02 mm	49.96 mm	−0.12%	47.30 mm	−5.32%	51.53 mm	+0.30%
	Depth	30.02 mm	29.96 mm	−0.20%	29.91 mm	−0.37%	29.85 mm	−0.57%
Rectangu-lar groove	Length	39.99 mm	40.01 mm	−0.05%	38.01 mm	−4.95%	41.17 mm	+2.95%
	Width	80.03 mm	80.01 mm	−0.02%	79.99 mm	−0.05%	80.01 mm	−0.02%
	Depth	20.01 mm	19.99 mm	−0.10%	19.86 mm	−0.75%	19.85 mm	0.80%

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
