# Peer review of "A Calibration Method for a Self-Rotating, Linear-Structured-Light Scanning, Three-Dimensional Reconstruction System Based on Plane Constraints"

_sensors, 2021, doi:10.3390/s21248359_

Round 1

Reviewer 1 Report

The authors present a method of calibrating a self-rotating scanning apparatus using the geometry of the measured device to to calculate constrained geometric parameters which aid in accurate measurement of the dimensions of the measured device.

The paper is overall very compelling and of interest. I believe this work is worthy of publication after some basic edits and additions. I believe the language could benefit from being improved - some sentences require multiple readings to understand. Some comments and questions are as follows:

  • Line 25 & line 43: First use of LSL and LIDRA acronyms are not explicitly defined.
  • The system might seem to be limited to measuring geometries which are symmetric. Is this the case? If so, (or not), can the authors add a statement to this fact.
  • Similarly, what constraints are there on the alignment of the central rotation axis with the centre of the measured device?
  • Does the orientation of the measured device have any impact? For example, if the top of the device is tilted forward, does this have any impact?
  • Section 4: The device measured by the authors is referred to as the "standard part". Perhaps this could be called "test piece", or similar?
  • Table 2: Here, a comparison is made for values of M & N and the resulting dimensions. How many scans were conducted to achieve these results? In line 166-167, it is stated this is done "multiple times", but the actual value(s) would be a good addition.
  • Table 2: An additional '-' in "Rectangular".
  •  

Author Response

Dear Editor and reviewers,

We gratefully thank the editor and all reviewers for their time spend making their constructive remarks and useful suggestions, which has significantly raised the quality of the manuscript and has enable us to improve the manuscript. Each suggested revision and comment, brought forward by reviewers was accurately incorporated and considered. Below the comments of the reviewers are response point by point, and here we did not list the changes but marked in red in revised paper.

Reviewer 1

  1. Line 25 & line 43: First use of LSL and LIDAR acronyms are not explicitly defined.

Reply: What you said is quite reasonable, we will add the explicitly defined when first used of LSL and LIDAR.

  1. The system might seem to be limited to measuring geometries which are symmetric. Is this the case? If so, (or not), can the authors add a statement to this fact.

Reply: We totally understand the reviewer’s concern. But the system is not limited to measuring geometries which are symmetric. And we will add a statement to this fact.

  1. Similarly, what constraints are there on the alignment of the central rotation axis with the centre of the measured device?

Reply: We totally understand the reviewer’s concern. There is no constraint on the alignment of the central rotation axis with the centre of the measured device.

  1. Does the orientation of the measured device have any impact? For example, if the top of the device is tilted forward, does this have any impact?

Reply: The orientation of the measured device do not have any impact,in fact, the top of the device can tilted forward or back.

  1. Section 4: The device measured by the authors is referred to as the "standard part". Perhaps this could be called "test piece", or similar?

Reply: We gratefully appreciate for your valuable comment. We will change the “standard part ” to “test piece”.

  1. Table 2: Here, a comparison is made for values of M & N and the resulting dimensions. How many scans were conducted to achieve these results? In line 166-167, it is stated this is done "multiple times", but the actual value(s) would be a good addition.

Reply: We appreciate for your valuable comment. There are 10 times scans were conducted to achieve these results. And the “multiple times” will be changed to “three times”.

  1. Table 2: An additional '-' in "Rectangular".

Reply: We are very sorry for our incorrect writing. We will delete the ‘-’.

Reviewer 2 Report

The following Comment should address before Publications:

  1. In the Transformation between coordinate systems sections, many mathematics equations are already published many papers, kindly use corresponding citations for that equations.
  2. In Transformation between coordinate systems, what is the notation of Suffix "r" in your paper, kindly give a proper explanation and need of three coordinates systems.
  3. There are very few related works that are discussed, kindly include many recent papers for this case.
  4. There is no comparison with previous published work like "Plane-constraint-based calibration method for a galvanometric laser scanner", etc.,
  5. Kindly elaborate on the novelty of your works. Only few things are included.
  6. the author discussed point cloud data, but LiDAR will give better object shapes. kindly compare it accordingly.

Author Response

Dear Editor and reviewers,

We gratefully thank the editor and all reviewers for their time spend making their constructive remarks and useful suggestions, which has significantly raised the quality of the manuscript and has enable us to improve the manuscript. Each suggested revision and comment, brought forward by reviewers was accurately incorporated and considered. Below the comments of the reviewers are response point by point, and here we did not list the changes but marked in red in revised paper.

Reviewer 2

  1. In the Transformation between coordinate systems sections, many mathematics equations are already published many papers, kindly use corresponding citations for that equations.

Reply: We appreciate for your valuable comment. We will add the corresponding citations for that equations.

  1. In Transformation between coordinate systems, what is the notation of Suffix "r" in your paper, kindly give a proper explanation and need of three coordinates systems.

Reply: We totally understand the reviewer’s concern. is the coordinate system of the self-rotating scanning LSL 3D reconstruction system, ”r” means the rotation centre. And we will give a proper explanation and need of three coordinates systems.

  1. There are very few related works that are discussed, kindly include many recent papers for this case.

Reply: We gratefully appreciate for your valuable comment. We will discuss more related works, such as:

Chi S, Xie Z and Chen W. A laser line auto-scanning system for underwater 3D reconstruction [J]. Sensors 2016; 16:1534.

Li L and Xi J. Free and global pose calibration of a rotating laser monocular vision sensor for robotic 3D measurement-system. In: International conference on optics in precision engineering and nanotechnology (icOPEN2013),Singapore, 22 June 2013, 87690I.

Xiao J, Hu X, Lu W, et al. A new three-dimensional laser scanner design and its performance analysis. Optik 2015;126: 701–707.

Manakov A, Seidel HP and Ihrke I. A mathematical model and calibration procedure for galvanometric laser scanning systems. In: Vision, modeling, and visualization workshop 2011, Berlin, Germany, 4–6 October 2011,pp.207–214.

Wissel T, Wagner B, Stuber P, et al. Data-driven learning for calibrating galvanometric laser scanners [J]. IEEE Sens J 2015; 15: 5709–5717.

Mao Y, Zeng L.C , Jiang J , et al. Plane-constraint-based calibration method for a galvanometric laser scanner[J]. Advances in Mechanical Engineering, 2018, 10(5):168781401877367.

  1. There is no comparison with previous published work like "Plane-constraint-based calibration method for a galvanometric laser scanner", etc.,

Reply: Thank you for your suggestion. We will add the comparison with previous published works such as "Plane-constraint-based calibration method for a galvanometric laser scanner" and “A new three-dimensional laser scanner design and its performance analysis” etc.

  1. Kindly elaborate on the novelty of your works. Only few things are included.

Reply: This paper proposes a calibration method for self-rotating scanning linear structured light (LSL) three-dimensional reconstruction system based on plane constraints. It can be quickly and accurately calibrated the position parameters between the coordinate system of LSL module and the coordinate system of the self-rotating scanning LSL 3D reconstruction system.

  1. the author discussed point cloud data, but LiDAR will give better object shapes. kindly compare it accordingly.

Reply: LiDAR is more suitable for large area and low precision scan. The self-rotating scanning linear structured light is suitable for medium or small area such as pipelines, tunnels, tanks or indoor environments, nuclear reactor internals, and underwater environments etc.

Round 2

Reviewer 2 Report

Kindly check grammatical errors

Author Response

Thank you for your advise. The grammatical errors has been corrected.